# Forecasting monthly residential natural gas demand in two cities of Turkey using just-in-time-learning modeling

Burak Alakent[1], Erkan Isikli[2]*, Cigdem Kadaifci[2], Tonguc S. Taspinar[3]

1 Department of Chemical Engineering, Bogazici University, Istanbul, Turkey, 2 Department of Industrial Engineering, Faculty of Management, Istanbul Technical University, Istanbul, Turkey, 3 SOCAR Türkiye, Vadistanbul Bulvar, Ayazağa Mahallesi, Azerbaycan Caddesi, Istanbul, Turkey

* isiklie@itu.edu.tr

## Abstract

Natural gas (NG) is relatively a clean source of energy, particularly compared to fossil fuels, and worldwide consumption of NG has been increasing almost linearly in the last two decades. A similar trend can also be seen in Turkey, while another similarity is the high dependence on imports for the continuous NG supply. It is crucial to accurately forecast future NG demand (NGD) in Turkey, especially, for import contracts; in this respect, forecasts of monthly NGD for the following year are of utmost importance. In the current study, the historical monthly NG consumption data between 2014 and 2024 provided by SOCAR, the local residential NG distribution company for two cities in Turkey, Bursa and Kayseri, was used to determine out-of-sample monthly NGD forecasts for a period of one year and nine months using various time series models, including SARIMA and ETS models, and a novel proposed machine learning method. The proposed method, named Just-in-Time-Learning-Gaussian Process Regression (JITL-GPR), uses a novel feature representation for the past NG demand values; instead of using past demand values as column-wise separate features, they are placed on a two-dimensional (2-D) grid of year-month values. For each test point, a kernel function, tailored for the NGD predictions, is used in GPR to predict the query point. Since a model is constructed separately for each test point, the proposed method is, indeed, an example of JITL. The JITL-GPR method is easy to use and optimize, and offers a reduction in forecast errors compared to traditional time series methods and a state-of-the-art combination model; therefore, it is a promising tool for NGD forecasting in similar settings.

## 1. Introduction

In the last few decades, there has been a shift in energy sources from fossil fuels to cleaner energy sources, such as wind and solar energy, mainly due to environmental concerns and related government regulations. However, these latter sources are

**Data availability statement:** A portion of the data underlying the results presented in the study are available from https://github.com/Balakent/NGD-in-Two-Cities.

**Funding:** The author(s) received no specific funding for this work.

**Competing interests:** The authors have declared that no competing interests exist.

dependent on weather conditions and require integration with grid technologies for continuous power generation. Natural gas (NG), typically, consists of (up to) ~95% of methane and 2–2.5% ethane-hexane + , with the remainder consisting of nitrogen, $CO_2$, oxygen and hydrogen, making NG closer to the cleaner side of the energy spectrum than fossil fuels. NG power plants are easy to build and highly reliable, making them invaluable for "clean" energy production. On the other hand, most countries depend on imports to maintain their NG supplies, and there is a delicate balance between imports and domestic demand. Storing excess imported gas above actual demand is difficult and would result in economic losses, while importing less than actual demand could result in a nationwide shortage. Therefore, accurate NGD predictions (NGDPs) are of utmost importance.

NGDP studies can be grouped by application area, which can vary from the whole world to cities, or by forecast horizon, which can be divided into (i) short-term, i.e., hourly to daily basis, (ii) medium-term, i.e., weekly to monthly basis, (iii) long-term, i.e., annual to decade basis, (iv) multi-term forecasting methods [1]. The focus in the literature has shifted from annual to daily forecasting methods from the 1970s to the present, possibly in line with advances in machine learning and forecasting methods; however, the share of medium-term studies in the entire NGDP literature has decreased from ~20% to ~15% during the same period [2], and the number of studies generating monthly forecasts is rather limited [3,4]. This may, in fact, be due to the dominant factors at different forecast horizons; monthly to quarterly forecasts are mostly and almost equally affected by temperature and economy [2], and both variables are difficult to predict for monthly periods [4,5].

## 1.1. A brief survey of NGDP models

Here, we adopt the perspective proposed by a review study [2] for the analysis of the historical evolution of the NGPD literature. Early studies on short-time NGDP in the 1960s focused on using statistical models with weather-related variables as features, while recognizing the time-series nature, particularly the seasonality of the data, and suggesting the inclusion of lagged input and output variables in the models [6,7]. The influence of time series models, proposed by Box and Jenkins [8], on NGDP became particularly evident during the 1970s and 1980s. For medium-term forecasting, a linear regression model consisting of heating degree days (HDD), cooling degree days (CDD), price of NG (PoG), number of customers (NoC), and NGD with a 12-month lag as inputs was used to obtain monthly NGDPs for residential customers in the US [9]. Monthly NGD for Taiwan was modeled using nonstationary seasonal autoregressive integrated moving average exogenous input (SARIMAX) models with lagged terms of temperature and PoG [10].

The priority given to nonlinear models increased by the late 1990s, possibly as a result of the advent of statistical learning methods [2]. Monthly NGDPs for Belgium were performed using an artificial neural network (ANN) model including exogenous features such as temperature, oil price and NoC [11]. The generation of one-to-three-day-ahead NG load forecasts was achieved by using a combination of ANN models, which incorporated the previous output, temperature, and wind speed as features.

The integration of learners and the updating of learner weights during forecasting were shown to enhance prediction accuracy [12]. Similarly, combination of ANN models was shown to provide more accurate daily-to-weekly forecasts of NGD in Poland, compared to linear and quadratic regression models [13]. In another study, daily NGD loads for a one-month period were forecasted using two years of historical data in a Support Vector Regression (SVR) model, in which daily temperature forecasts during the test period were taken as simple averages of those of the previous two years during the same period [14]. In short-term NGD forecasting, Gil and Deferrari used a semi-empirical model to describe the relation of NGD with the number of users and temperature for short-term forecasts for Argentina [15].

In the last ~15 years, statistical models, time series methods and machine learning tools have been combined in coherent frameworks for accurate NGDPs. In short-term NGD forecasting, features related to weather, number of subscribers, and previous demand values have often been used to train learners, such as Linear Regression, Local SVR, Gradient Boosting (GB), and ANN models [16–18]. In recent studies, special attention has been given to the "data leakage" problem, especially related to weather-related variables; i.e., forecasts of exogenous variables are used in models for predicting future NGD values. For instance, one hour-to-60 hours ahead forecasts of temperature have been used in a model consisting of temperature-independent and temperature-dependent terms [19], and in a long short-term memory (LSTM) ANN model [20]. More recently, various deep learning architectures, such as the combination of convolutional NN (CNN) with attention modules to extract short-time relations, and LSTM models to capture longer-time relations, have been tailored for NGDP of various cities in China using training sets of moderate size, e.g., ~200–350 observations [21,22].

Among the few medium-term forecasting studies, mutual information on the exogenous variables, such as population, gross domestic product (GDP), PoG, and HDD, was used to select a subset of features to be incorporated into local linear neuro-fuzzy learner and Hodrick–Prescott filter to determine U.S. NGDPs for a 12-month period [23]. Extreme Learning Machine (ELM) and ANN models were shown to yield accurate monthly residential NGDPs for Karaj [24] and Tehran [25] in Iran, using weather-related variables as exogenous features; however, these studies used actual weather data during the test period, which makes the models difficult to use in practice.

## 1.2. NGDP studies for Turkey

Demand for NG in Turkey is constantly increasing, and residential sector has accounted for >35% of its domestic NG consumption in 2023 [26]. While Turkey has been investing on NG resources, a large portion of Turkey's source of NG is imports from Russia, Azerbaijan, Iran via the pipelines, and shipped Liquefied NG from Nigeria and Algeria [27]. NGDP for the entire country and individual cities in Turkey has been studied for the last ~25 years using various statistical and machine learning methods. One-day-ahead forecasts were conducted using SARIMAX and ANN models, and moving window linear regression model for Sakarya [28,29], multivariate adaptive regression splines (MARS), Lasso and Ridge Regression for Ankara [30–32], Fourier Series for Istanbul, Ankara and Eskisehir [33], SARIMAX, nonlinear ARX (NARX), ANN and LSTM models for Turkey [34], and ANN model for Istanbul [35]. The input features in these studies usually consisted of past demand values, instantaneous weather-related variables, the number of gas users, exchange rates and other economic variables.

In one of the earliest medium-term NGDP studies in Turkey, regression models, including HDD and previous prediction residuals as input features, were shown to improve monthly prediction accuracy compared to a single model for Eskisehir [36]. NGD in Istanbul was forecasted for a seven-month test period using ANN models, incorporating separately coded months, NoC and monthly average customers [37]. Relation of monthly NGD in Turkey and meteorological variables was examined using metaheuristic algorithms on additive regression models [38]. In a long-term NGDP analysis of Turkey spanning from 2005 to 2023, employing different base temperature scenarios, it was predicted that a 1$^{\circ}$C change in the HDD base temperature would result in ~10% change in NG consumption [39]. In another study, daily NGDPs were modeled using HDD, exchange rate, and PoG as features in linear regression models. These models were developed for the purpose of annual forecasting, with the understanding that they may be applicable under stable and unstable economy

scenarios [40]. Subset regression methods were used to select features among gross national product (GNP), population, and growth rate [41], while simulated annealing (SA) [42], hybrid SA+genetic algorithm (GA) [43], artificial bee colony algorithm [44], and cause-effect relationship-based Grey-prediction model [45] were used for forecasting Turkey's NGD under different scenarios. In contrast to scenario-based forecasts, long-term (>15 years) point predictions of annual NGD were obtained via a high-order ARI model [46], and modified logistic and linear equations, involving the (forecasted) population as the single predictor [47]. In a recent study, the fractional non-linear grey Bernoulli model, which was optimized by grey wolf optimization, was shown to yield better one-year ahead forecasting results compared to linear regression and ARIMA models using a limited dataset of only 14 years [48]. A synopsis of NGDM studies conducted in Turkey is provided in the Appendix.

### 1.3. NGDP studies focusing on medium-term forecasting horizon

The extant studies have notable value for policy decisions and design purposes; however, most of them, particularly those pertaining to Turkey, use exogenous-variable models. In these models, either the actual (realized) values of the input features were used during the test period (ex-post forecasts), or the details of how input features were obtained during the test period were discussed superficially. One-day ahead estimations of variables related to weather might be helpful for short-term forecasting; however, forecasts of weather-related variables for longer than a week period would be less reliable, thereby decreasing the accuracy of exogenous variable models for monthly NGDP [15,49]. Scenario-based approaches, which use a palette of different forecast sets for future values of input features, have proven effective in long-term forecasting. However, the low-frequency change of exogenous variables, such as GDP, coupled with the challenging nature of accurately predicting variables like the inflation rate, poses a significant challenge in obtaining reliable monthly point NGDPs [50]. Consequently, methods employed for monthly forecasting (ex-ante forecasts) are rather limited in number and can be classified into two groups: (i) models that use the forecasted values of exogenous variables for future points and (ii) models that use previous NGD values, along with the possible inclusion of calendar information.

As an example of the former approach, Gil and Deferrari expanded their daily temperature predictions to monthly predictions by assuming a normal distribution of daily temperatures around a monthly mean, approximated by a simple periodic function within a year [15]. In a similar study, where one-day and one-month ahead residential NGD in Turkey were predicted using linear regression models with HDD, CDD, price and holiday effects as input features, daily temperature was estimated using Ornstein-Uhlenbeck stochastic process models [51]. A nonlinear model including the predicted seasonal temperature effect was used to determine monthly NGDP in the Czech Republic [52].

The second approach entailed the integration of linear Hinges models to describe the seasonal behavior, augmented by a trend component, to determine monthly NGDP in Spain. The incorporation of explanatory variables proved instrumental in elucidating weekly effects [5]. ANN and adaptive neuro-fuzzy inference system (ANFIS) models using only previous NGD values without auxiliary features were shown to yield higher prediction accuracy compared to ARIMA model in forecasting NGD over a 15-week-period in Turkey [53]. In a study on forecasting monthly NG consumption for Istanbul, Turkey, over a one-year period, the authors used a set of features comprising the seasonal index, NG consumption at lag 12 (month), temperature, city population, and PoG [54]. NGDPs using an adjusted seasonal grey model over a four-year period [55], and Holt-Winters exponential smoothing and SARIMAX models for monthly periods [56], both yielded acceptable forecast accuracy for Turkey. Nonlinear grey Bernoulli model based on Hodrick-Prescott (HP) filter was shown to yield the smallest prediction error among other seasonal models in predicting monthly NGD values of European Union countries [4]. A comparison of ARIMA, NARNN, SVR, and LSTM models using the weekly NGD in Turkey during the COVID-19 pandemic, was conducted in a recent study for one-week ahead forecasts [57]. A study that combined CNN and LSTM models for forecasting daily NGDPs in Chinese cities used only previous NGD values for model training, demonstrating that deep learning architectures can also be used for medium-term NGDPs under a similar setting [21]. Quarterly NG consumption data in China was analyzed using moving average and seasonal indices to yield NG

consumption forecasts until 2035 [58]. A recent study on an ensemble of ETS and SARIMA models, obtained via resampling the remainder of Seasonal-Trend decomposition using LOESS (STL) on NGD for 18 European countries, yielded a good prediction accuracy for a 12-month ahead period [3]. The Salp swarm algorithm, in conjunction with Extreme Gradient Boosting (XGB), was employed to forecast five-month ahead NGDPs for the UK and the Netherlands within a moving window (MW) frame [59].

### 1.4. Perspective of the present study

The monthly NGD data of Bursa and Kayseri, two cities in Turkey, from 2014 to 2024 were provided by The State Oil Company of Azerbaijan Republic (SOCAR), an Azerbaijani local NG distribution company employed in Turkey. SOCAR was established in 1992 with the mission of managing the nation's oil and natural gas industry. This globally recognized energy giant plays a pivotal role in delivering the abundant energy resources of the Caspian Sea to global markets. SOCAR's activities span a broad range, including: the exploration and operation of Azerbaijan's oil and natural gas reserves through the use of advanced technologies. As of 2023, SOCAR's production from reserves in the Caspian Sea amounted to approximately 34 million tons of oil and 35 billion cubic meters of natural gas annually. SOCAR's role in the refining of oil and the production of energy products is also noteworthy. The company's *Başkal* and *Yeni Heydar Aliyev* refineries possess a combined processing capacity of 7 million tons of crude oil per year. Furthermore, SOCAR has strategically invested in transportation infrastructure to facilitate the delivery of energy products to domestic and international markets, establishing a robust logistics network comprising pipelines and terminals. Approximately 50 million tons of oil are transported annually via the *Baku-Tbilisi-Ceyhan Pipeline*. Additionally, SOCAR Turkey Research, Development and Innovation Inc. (SOCAR Turkey R&D), established in 2019, develops innovative, sustainable, environmentally friendly, and market-oriented products and digital technologies, and provides R&D services to all its stakeholders with its deep-rooted experience in the energy sector. SOCAR has invested in various projects in Turkey, Georgia, and other countries to be among the global players in the energy sector. Notably, SOCAR has augmented its regional effectiveness through substantial investments, amounting to a total of 18.5 billion dollars. These investments have been made through initiatives such as the acquisition of PETKİM in Turkey and its subsequent integration with the recently established STAR Refinery, the completion of the Trans Anatolian Natural Gas Pipeline (TANAP) Project, and the acquisition and operation of natural gas distribution companies such as Bursagaz and Kayserigaz.

To comply with the specifications of the import agreements, it is necessary to make monthly NGD forecasts for the upcoming year. In the present study, a novel ex-ante NGDP method, namely Just-In-Time-Learning-Gaussian Process Regression (JITL-GPR), is proposed. The JITL-GPR model uses only the previous NGD values and calendar data, offering a more nuanced interpretation from an online learning perspective as opposed to batch learning. In batch learning, a constant model (or an ensemble of models) is constructed from a given (historical) dataset, and the response variable value of a new query point is predicted using this model. Conversely, in online learning, a model is continuously adapted or changed upon varying operating conditions [60–62]. The accuracy of a constant model relating inputs to outputs, derived using batch learning, is likely to degrade over time; therefore, relaxing the constant model assumption is an advantage of online learning [63]. Indeed, online learning methods have already been successfully used (rather implicitly) for NGDP: Linear learning methods were found to demonstrate enhanced efficacy when used simultaneously with recursive [16], MW [1,19,59], or nearest neighbor (NN) [17] methods. Furthermore, the prediction accuracy of nonlinear learning methods was shown to increase when trained in an adaptive manner [12].

The novelties of JITL-GPR are as follows: (i) Instead of extracting trend-seasonality information directly and globally from the entire data, as most of the time series methods do, this information is captured indirectly through the selection of a convenient "window" of observations on a two-dimensional grid of the historical year-month plane. (ii) A convenient kernel is designed to exploit the trend-seasonality information to be used in GPR. (iii) The procedure is repeated for each query point, akin to Just-In-Time-Learning (JITL) or local learning [64], while recent information is also included, similar

to a MW method [65,66]. Hence, the method yields predictions consistent with previous years, while adapting to changes in recent months. In comparison to conventional time series methodologies such as SARIMA and ETS, JITL-GPR was shown to exhibit superior accuracy in terms of NGDP estimation.

Section 2 encompasses a concise overview of Time Series Methods, GPR and JITL. Following the presentation of the data's specifics, a comprehensive discourse on the results obtained using time series methods and JITL-GPR is provided in Section 4. The final section proposes several future research directions.

## 2. Theoretical background

### 2.1. Time series modeling

**2.1.1. Seasonal and trend decomposition using LOESS (STL).** The pattern of a time series can be decomposed into sub-patterns in many cases, thereby defining the various components of the series individually. This type of decomposition frequently enhances the accuracy of forecasting by facilitating a more profound comprehension of the series' behavior [67]. Consequently, decomposition of a time series is a useful instrument for the analysis of a series and the subsequent selection of a method for its modeling. In the present study, we employed the STL (seasonal-trend decomposition) procedure that consists of an inner loop nested inside an outer loop. Developed by [68], this non-parametric additive decomposition method provides a simple design based on local regression smoothing (LOESS) that is flexible in specifying the amounts of variation in the trend and seasonal components. It is widely regarded as a valuable tool for elucidating the underlying patterns in datasets that exhibit substantial seasonal effects. A notable strength of STL lies in its capacity to handle series with missing values, yielding robust trend and seasonal components that are not distorted by transient, aberrant behavior in the data [68]. In other words, it can accommodate non-fixed seasonal patterns and handle non-linear trends in the presence of outliers. The general form of the STL model can be expressed as follows in Equation 1:

$$y_t = (\alpha + \beta t) + S_t + \varepsilon_t = \hat{y}_t + \varepsilon_t \tag{1}$$

where $y_t$ is the actual value of the time series at time point $t$; $\alpha + \beta t$ corresponds to the trend component; $S_t$ is the seasonal index at time point $t$; $\varepsilon_t$ is the error component at time point $t$. While STL can manage any kind of seasonality (i.e., weekly, monthly, or quarterly), allowing users to determine the trend's smoothness and enable the seasonal component to alter over time, it only offers additive decomposition capabilities and does not automatically handle calendar or trading day variations [69]. It can be implemented easily since it relies on numerical methods rather than mathematical modeling [70].

**2.1.2. Seasonal autoregressive integrated moving average (SARIMA).** Along with smoothing methods, ARIMA is one of the most widely used and effective time series modeling approaches. It aims to describe the behavior of a time series based on autocorrelations [71]. This type of models posits that future values of a time series are generated from a linear function of its past observations and some white noise errors. The model is specified by three parameters: $p$, the order of the autoregressive component; $d$, the order of the differencing required for the series to be mean stationary; and $q$, the order of the moving average component. Equation 2 illustrates an ARIMA($p$, $d$, $q$) model:

$$\Delta^d y_t = \mu + \varphi_1 \Delta^d y_{t-1} + \varphi_2 \Delta^d y_{t-2} + \ldots + \varphi_p \Delta^d y_{t-p} - \theta_1 \varepsilon_{t-1} - \theta_2 \varepsilon_{t-2} - \ldots - \theta_q \varepsilon_{t-q} \tag{2}$$

where $\Delta^d y_t$ is the $d$-th difference of the original series, $\varphi_i$ is the parameter for the $i^{th}$ autoregressive term, $\theta_j$ is the parameter for the $j^{th}$ moving average term, and $i = \overline{1, p}$ and $j = \overline{1, q}$. For a comprehensive exposition on the underpinnings of ARIMA-type models, readers are directed to the seminal contributions of [72]. The ARIMA methodology was extended to incorporate seasonal trends along with primary stochastic trend to facilitate the modeling of time series characterized by periodicity and regular patterns across diverse time scales. Known as SARIMA, this type of model is denoted

by $ARIMA(p, d, q)(P, D, Q)_s$, where $D$ denotes the degree of seasonal differencing, $P$ denotes the number of seasonal autoregressive components, $Q$ denotes the number of seasonal moving average components, and $s$ denotes the seasonal cycle (e.g., for monthly data with yearly seasonality, $s = 12$). Expressing a SARIMA model explicitly is tedious; therefore, lag polynomials are most often used. A scrambled mathematical representation of an $ARIMA(0, 1, 1)(0, 1, 1)_{12}$ model is formulated in Equation 3 below as an example:

$$\Delta_{12}^D \Delta^d y_t = \mu + \Theta_1 \theta_1 \varepsilon_{t-13} - \Theta_1 \varepsilon_{t-12} - \theta_1 \varepsilon_{t-1} + \varepsilon_t \tag{3}$$

where $\Delta_{12}^D y_t$ is the $D$-th seasonal difference of the original series and $\Theta_1$ is the parameter for the seasonal moving average component.

### 2.1.3. Exponential smoothing state space models (ETS).

State space models, first introduced in the late 1970s, offer a high degree of flexibility in the specification of parameters for level, trend, and seasonal components. This is primarily achieved through the use of two equations: the observation equation and the state equation [73]. As intrinsically non-parametric models, they are reliable and computationally efficient. For a more in-depth exploration of this topic, readers are directed to [74]. A notable distinction of state space models for exponential smoothing is the inclusion of an error component, in addition to the trend and seasonal components, which distinguishes them from the smoothing methods previously introduced in the related literature. This class of models is commonly denoted by the acronym $ETS(x, y, z)$, where $x$ designates whether errors are additive (A) or multiplicative (M), $y$ indicates the trend is additive or multiplicative nature, and $z$ denotes the seasonality is additive or multiplicative character. To illustrate, the following two representations of the ETS(M,A,N) model are provided below. The general representation is given in Equations 4–6, whereas the state space representation is given in Equations 7–8. This is somewhat equivalent to Holt's linear method, but the errors in this case are multiplicative rather than additive.

$$\text{Forecast: } \hat{y}_t = (L_{t-1} + B_{t-1})(1 + \varepsilon_t) \tag{4}$$

$$\text{Level : } L_t = (1 + \alpha \varepsilon_t)(L_{t-1} + B_{t-1}) \tag{5}$$

$$\text{Trend : } B_t = \alpha \beta \left( L_{t-1} + B_{t-1} \right) \varepsilon_t + B_{t-1} \tag{6}$$

$$\text{Obs. Equation : } y_t = \begin{bmatrix} 1 \\ 1 \end{bmatrix}^T \boldsymbol{x}_{t-1}(1 + \varepsilon_t) \tag{7}$$

$$\text{State Equation : } \boldsymbol{x}_t = \begin{bmatrix} 1 & 1 \\ 0 & 1 \end{bmatrix} \boldsymbol{x}_{t-1} + \begin{bmatrix} 1 \\ 1 \end{bmatrix}^T \boldsymbol{x}_{t-1} \begin{bmatrix} \alpha \\ \beta \end{bmatrix} \varepsilon_t \tag{8}$$

where $\hat{y}_t = E[y_t | y_{t-1}, y_{t-2}, \dots, y_1]$ is the one-step-ahead forecast, $L_t$ denotes the baseline value of the series at time point $t$, $\boldsymbol{x}_t = \begin{bmatrix} L_t \\ B_t \end{bmatrix}$ is the state vector and the one-step ahead forecast errors (estimation data), $\varepsilon_t = \frac{y_t - (L_{t-1} + B_{t-1})}{L_{t-1} + B_{t-1}}$, follow a normal distribution with zero mean and constant variance. The selection of the ETS model is contingent upon the characteristics of the time series, and estimation of model parameters is frequently executed through the Maximum Likelihood Estimation (MLE) technique. This approach involves maximizing the "likelihood" and thereby minimizing the sum of squared errors while optimizing the parameters. Table 1 presents a selection of the most commonly used ETS methods, along with their

**Table 1. Commonly used ETS models.**

| Method | ETS Notation | Reference |
|---|---|---|
| Simple Exponential Smoothing (SES) | ETS(A,N,N) | [75] |
| Holt's Linear Trend method | ETS(A,A,N) | [76] |
| Holt-Winters' Additive Method | ETS(A,A,A) | [77] |
| Holt-Winters' Multiplicative Method | ETS(M,A,M) | [77] |
| Holt-Winters' Damped Method | ETS(A,A$_d$,A) | [78] |

respective notations and references. A comprehensive overview of the equations in state space for every model in the ETS framework can be found in [69].

**2.1.4. Trigonometric box-cox transform, ARMA errors, trend, and seasonal components (TBATS).** Integrating the principles of ETS and other methodologies, TBATS handle multi-frequent seasonality and non-linear trends. TBATS was introduced by [79] as an extension of ETS to model multiple seasonal patterns in a time series simultaneously. In the abbreviation TBATS, the first letter of each word indicates a particular aspect of the model: B = Box-Cox transformation, A = ARMA errors, T = Trend, and S = Seasonality. The first T indicates that a trigonometric formulation was adopted to model seasonality. The method employs a Box-Cox transformation to stabilize variance and incorporates ARMA components to enhance the accuracy of error modeling. For a detailed exposition of TBATS, readers are directed to consult [80].

**2.1.5. The aggregated forecast through exponential reweighting (AFTER).** This simple and efficient method was initially proposed by [81] as a means of combining heterogeneous forecast models. Subsequent modifications have been made [82], and the method has been applied to various cases [83] since its initial introduction. Considering individual model performance history and focusing on minimizing prediction error over time, AFTER adjusts the combination weights, which undergo an exponential decay that is based on the performance of the models. This approach stands in contrast to classical forecast combination methods, which are limited to fixed or predetermined weights. For a more detailed description of AFTER, the reader is referred to [84] and [85].

## 2.2. Gaussian process regression (GPR)

Bayesian linear regression (BLR) is a parametric model, consisting of basis functions evaluated at finite training points. Given a set of $N$ data points $\{(\boldsymbol{x}_1, y_1), (\boldsymbol{x}_2, y_2) \ldots (\boldsymbol{x}_N, y_N)\}$ with $\boldsymbol{x}_n \in \mathbb{R}^p$ and $y_n \in \mathbb{R}$, the response variable $y_n$ is assumed to be normally distributed about the regression function, which is determined by the product of a basis function $\varphi(\cdot)$ evaluated at $\boldsymbol{x}_n$ and a parameter vector $\boldsymbol{w} \in \mathbb{R}^p$. This model is augmented with a Gaussian prior, specified by a zero mean and an identity covariance matrix:

$$\boldsymbol{w} \sim N\left(\boldsymbol{0}_p, \ \sigma_w^2 \boldsymbol{I}_p\right), \ y_n|\boldsymbol{x}_n = N(\varphi(\boldsymbol{x}_n)^T \boldsymbol{w}, \sigma_\varepsilon^2) \tag{9}$$

In the above equation, $\boldsymbol{0}_p$ and $\boldsymbol{I}_p$ represent the zeros vector and identity matrix, respectively, each with $p$ elements. Meanwhile, $\sigma_w^2$ and $\sigma_\varepsilon^2$ represent the variance terms for the prior of $\boldsymbol{w}$ and the independent additive noise, respectively. Defining the regression function $f(\boldsymbol{x}_n, \boldsymbol{w}) = E_y\{y|\boldsymbol{x}_n, \boldsymbol{w}\}$ yields:

$$f(\boldsymbol{x}_n, \boldsymbol{w}) = \varphi(\boldsymbol{x}_n)^T \boldsymbol{w} \tag{10}$$

$$\boldsymbol{f}(\boldsymbol{w}) = \left[\varphi(\boldsymbol{x}_1)^T \boldsymbol{w} \ \varphi(\boldsymbol{x}_2)^T \boldsymbol{w} \ \ldots \varphi(\boldsymbol{x}_N)^T \boldsymbol{w}\right]^T = \Phi(\boldsymbol{X})\boldsymbol{w} \tag{11}$$

In the equation above, $\boldsymbol{X} = \{\boldsymbol{x_1}\ \boldsymbol{x_2}, \ldots \boldsymbol{x_N}\}$ and $\Phi(\boldsymbol{X}) \in \mathbb{R}^{N \times p}$. Since $\boldsymbol{f}$ is a linear function of $\boldsymbol{w}$ with isotropic Gaussian prior, the following relations can be written [86]:

$$E_{\boldsymbol{w}}\{\boldsymbol{f}(\boldsymbol{w})\} = \boldsymbol{0_N} \tag{12}$$

$$\underset{\boldsymbol{w}}{cov}(\boldsymbol{f}(\boldsymbol{w})) = E_{\boldsymbol{w}}\left\{\boldsymbol{f}(\boldsymbol{w})\,\boldsymbol{f}(\boldsymbol{w})^{\boldsymbol{T}}\right\} = \sigma_w^2\,\Phi(\boldsymbol{X})\,\Phi(\boldsymbol{X})^{\boldsymbol{T}} \tag{13}$$

The inner product between the basis functions, as depicted in the equation above, suggests a kernel definition, such that $k(\boldsymbol{x}_u, \boldsymbol{x}_v) = \sigma_w^2\,\varphi(\boldsymbol{x}_u)^T\varphi(\boldsymbol{x}_v)$ for any two points $\boldsymbol{x}_u$ and $\boldsymbol{x}_v$ in the dataset. This yields $cov(\boldsymbol{f}) = \boldsymbol{K}(\boldsymbol{X})$, where $\boldsymbol{K}(\boldsymbol{X}) \in \mathbb{R}^{N \times N}$ is the Gram matrix. From the function-space perspective, any subset of random variables sampled from the GP $f(\boldsymbol{x})$ has a joint Gaussian distribution with a mean of $m(\boldsymbol{x})$, that has been taken to be equal to zero in the weight-space perspective discussed above, and a covariance function defined via a kernel $k\left(\boldsymbol{x}, \boldsymbol{x}'\right)$ [87]:

$$f(\boldsymbol{x}) \sim GP(m(\boldsymbol{x}), k\left(\boldsymbol{x}, \boldsymbol{x}'\right)) \tag{14}$$

In contrast to the BLR approach, the basis function in GPR is evaluated at an infinite number of points GPR, thereby yielding a nonparametric model. The resulting covariance relation shows that the change of the regression function at different points in feature space, i.e., the covariance, is completely determined by the kernel function defined in the original feature space. This implies that the behavior of the function will be correlated for "similar" points in the original space. Using $n = 1, 2, \ldots N$ for the training set and $n = q$ for a test point, the marginal distributions of $\boldsymbol{y} = [y_1\ y_2\ \ldots y_N]^T$ (also called the evidence function), $\boldsymbol{y}_{N+1} = [y_1\ y_2\ \ldots y_N\ y_q]^T$ and the conditional distribution of $y_q\,|\boldsymbol{y}$ may be determined using matrix normal distribution properties as follows:

$$p(\boldsymbol{y}|\boldsymbol{X}) = \int p(\boldsymbol{y}|\boldsymbol{f}, \boldsymbol{X})p(\boldsymbol{f}|\boldsymbol{X})\,d\boldsymbol{f} \sim N(\boldsymbol{0}_N, \boldsymbol{C}_N)\ \text{with}\ \boldsymbol{C}_N = \sigma_\varepsilon^2 \boldsymbol{I}_N + \boldsymbol{K}(\boldsymbol{X}) \tag{15}$$

$$p(\boldsymbol{y}_{N+1}|\boldsymbol{X}, x_q) = \int p(\boldsymbol{y}_{N+1}|\boldsymbol{f}, \boldsymbol{X}, x_q)p(\boldsymbol{f}|\boldsymbol{X}, x_q)\,d\boldsymbol{f} \sim N(\boldsymbol{0}_{N+1}, \boldsymbol{C}_{N+1}),$$

$$\text{with}\ \boldsymbol{C}_{N+1} = \begin{bmatrix} \boldsymbol{C}_N & k(\boldsymbol{X},\ x_q) \\ k(\boldsymbol{X},\ x_q)^T & \sigma_\varepsilon^2 + k(x_q, x_q) \end{bmatrix}_{(N+1) \times (N+1)} \tag{16}$$

$$y_q\,|\boldsymbol{y} \sim N(k(\boldsymbol{X},\ x_q)^T \boldsymbol{C}_N^{-1} \boldsymbol{y}, \sigma_\varepsilon^2 + k(x_q, x_q) - k(\boldsymbol{X},\ x_q)^T \boldsymbol{C}_N^{-1} k(\boldsymbol{X},\ x_q) \tag{17}$$

In the above formulation, $\boldsymbol{C}_N$ should be positive definite. As a result, the expected value of the response variable at $\boldsymbol{x}_q$ is a linear combination of response variables in the training set weighted with respect to the scaled kernel distance to the query point [86]:

$$E\{y_q\,|x_q, \boldsymbol{X}, \boldsymbol{y}\} = f(x_q\,|\boldsymbol{X}, \boldsymbol{y}) = k(\boldsymbol{X},\ x_q)^T \boldsymbol{C}_N^{-1}\,\boldsymbol{y} = \sum_{n=1}^{N} b_n k(\boldsymbol{x}_n,\ x_q)\,y_n \tag{18}$$

A typical example of a kernel function is the radial basis function (RBF) $k(\boldsymbol{x}_u, \boldsymbol{x}_v) = \sigma_f^2\,\exp\left(\frac{\|\boldsymbol{x}_u - \boldsymbol{x}_v\|_2^2}{2\sigma_l^2}\right)$, in which $\sigma_f^2$ and $\sigma_l^2$ denote the variance of the function (signal) and the squared length scale of the kernel, i.e., the range of the influence of the kernel function in the input space. It is possible to determine the optimum values of the kernel parameters, i.e.,

hyperparameters of the GPR. Using $\boldsymbol{\theta} = [\theta_1 \ \theta_2 \ldots \theta_k]$ as hyperparameters of GPR, e.g., $\boldsymbol{\theta} = [\sigma_f^2 \ \sigma_l^2]$ for the RBF kernel defined above, one may obtain (using Equation 15) the following log-likelihood:

$$\log p\left(\boldsymbol{y}|\theta,\boldsymbol{X}\right) = -\frac{1}{2}\left|\boldsymbol{C}_N\right| - \frac{1}{2}\ \boldsymbol{y}^T\boldsymbol{C}_N^{-1}\ \boldsymbol{y} - \frac{N}{2}\log 2\pi$$

(19)

The function above may be minimized using gradient-based techniques; however, initialization of the hyperparameters is important due to the non-convex nature of the log-likelihood function. Lastly, for a full Bayesian treatment, it is essential to define the priors of $\boldsymbol{\theta}$ and incorporate $\log p\left(\boldsymbol{\theta}\right)$ into $\log p\left(\boldsymbol{y}|\theta,\boldsymbol{X}\right)$.

### 2.3. Just-in-time-learning (JITL)

The development of JITL can be traced back to two seminal studies, in which local weighted regression and instance-based learning were proposed. Contrary to global models that seek to describe data across the entire domain using a single representation, local weighted regression was proposed for approximating nonlinear data over different regions [88]. Similarly, instance-based classification was based on the label of samples similar to the query point, instead of a single decision rule [89]. The accuracy of instance-based predictions for real-valued functions under realistic conditions was also theoretically justified [90]. These studies merged in JITL (or Lazy Learning) modeling, which has been employed in numerous modeling and process control applications [91–93], including time series analysis [64].

In the context of JITL modeling, the objective is to identify a subset of training points $\Phi$, that exhibits "similarity" to the given query point $\boldsymbol{x}_q$ consisting of the input features from the entire available historical dataset TS, such that $\Phi \subseteq TS$. The training of the learner $L(\cdot, \ \cdot)$ is performed on $\Phi$ with the aim of predicting the response for the query point $\boldsymbol{y}_q$, given by $\hat{\boldsymbol{y}}_q = L(\boldsymbol{x}_q, \ \Phi)$. This process is repeated for each query point, and the decision of whether to concatenate the predicted query point (along with its real or predicted response variable value) to TS is application-dependent. The key to obtaining accurate predictions via JITL lies in the procedure of selecting the relevant subset. Typically, proximity in feature space, as measured by Euclidian distance between feature values, has been used to determine the similarity of observations. The $k$-neighbors of each query point ($k$ usually determined by a validation set) are determined as the $k$-observations with the smallest Euclidian distance to the query point [91]. However, the efficacy of this criterion is contingent on the learner and the data environment. For instance, a (weighted) linear regression may be preferred over more advanced learning methods for small $k$ values, since it would be difficult to train the latter using a small sized $\Phi$ due to a presumable high capacity. As another example, MW modeling is usually used to adapt to the most recent operating conditions, and MW can be classified as a type of JITL method, in which relevant data is selected not in relation to proximity in feature space, but rather to proximity to sampling time. Consequently, various methods for instance selection have been proposed in the literature on soft-sensors [63,94–96]. In the present study, proximity on a 2-D grid comprised of years and months is adopted as the similarity measure. A relevant subset of the training set is selected to span $W_y$ years and $W_m$ months prior to the year and month of the query point, respectively (see Section 4.2 for details). In this context, the hyperparameters $W_y$ and $W_m$, which are instrumental in determining the size of the local training set, are tuned using rolling-origin forecasting in the training set, starting from a buffer period of 48 observations.

### 2.4. Metrics for assessing prediction accuracy

In the context of validating and selecting the most suitable models, as well as for parameter tuning (i.e., model fit), comparing alternative models, and evaluating the overall forecast performance [97], the most frequently used performance measures are the root mean square error (RMSE), the mean absolute error (MAE), and the mean absolute percentage error (MAPE).

Equations 20–22 present the formulae of these performance measures, where $y_i$ represents the actual values, $\hat{y}_i$ denotes the predicted values, and $n$ corresponds to the number of observations in the dataset. For a more in-depth exploration of additional performance measures, readers are encouraged to refer to the works of [98] and [99].

$$\text{MAE} = \frac{1}{n} \sum_{i=1}^{n} \left| y_i - \hat{y}_i \right|$$

(20)

$$RMSE = \sqrt{\frac{\sum_{i=1}^{n} \left( y_i - \hat{y}_i \right)^2}{n}}$$

(21)

$$\text{MAPE} = \frac{100}{n} \sum_{i=1}^{n} \left| \frac{y_i - \hat{y}_i}{y_i} \right|$$

(22)

## 3. Data description

The tariff methodology is instrumental in determining the revenues of natural gas distribution companies with regulated tariffs by calculating the System Usage Fee (SUF) based on consumption levels. A tariff period is set to last five years, during which SUF is calculated at the beginning based on past realizations using statistical models as follows: SUF = Revenue Requirement (OPEX + CAPEX)/ Total Consumption, where OPEX stands for operating expenses and CAPEX stands for capital expenditures.

Accurate natural gas consumption estimates are critical for regulated companies, as deviations in five-year estimates can lead to unexpected income losses or gains due to regulatory revisions. While OPEX and CAPEX transfers are mathematically predictable, external and environmental factors complicate long- and short-term consumption forecasts. Advanced modeling techniques and data analysis methods are used to improve these estimates, but monthly accuracy remains challenging. Enhancing natural gas consumption forecasts is strategically vital for efficient resource management in the natural gas distribution sector.

The existing dataset, provided by SOCAR, consists of monthly residential NGD consumption in million standard m$^3$ (msm$^3$) in Bursa and Kayseri from January 2014 to August 2024. This dataset was divided into a training/validation set of 108 observations (9 full years) between 2014 and 2022 and a test set of 19 observations (1 year and 7 months) between 2023 and 2024. The former set was used for building models and tuning the hyperparameters, while the latter was exclusively used for giving unbiased test set performance, similar to the monthly NGDP values for a one-year-period, as stipulated by the company. The NG consumption is ascertained through the monitoring of local sensors situated in residential areas and buildings. It was reported by SOCAR that the summer period readings, i.e., July, August and September in Bursa, and June, July, August and September in Kayseri, during 2014–2019, were not conducted exactly on time; the readings for the first two months were delayed, and the last months readings included some of the consumption of the prior months. Consequently, the Results section details the correction of summer period data in the training set prior to forecasting. The construction of time series and JITL-GPR models was then executed using the corrected training dataset. Figs 1 and 2 below depict the logarithmically transformed time series of NG consumption, in addition to its primary components, for Bursa and Kayseri, respectively. A visual examination of these plots indicates the presence of trend and additive seasonality in the datasets of both cities.

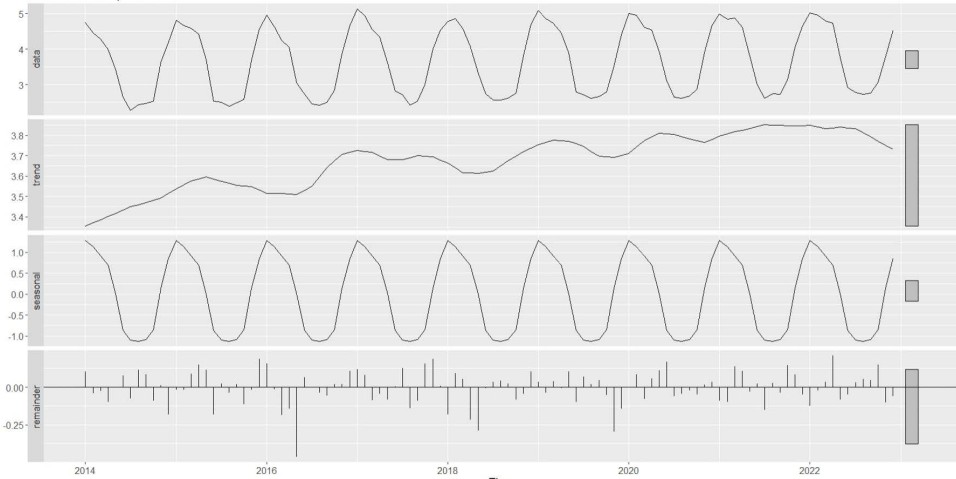

**Fig 1. The STL decomposition of the logarithm of Bursa's corrected NG consumption data (January 2014 to December 2022).**

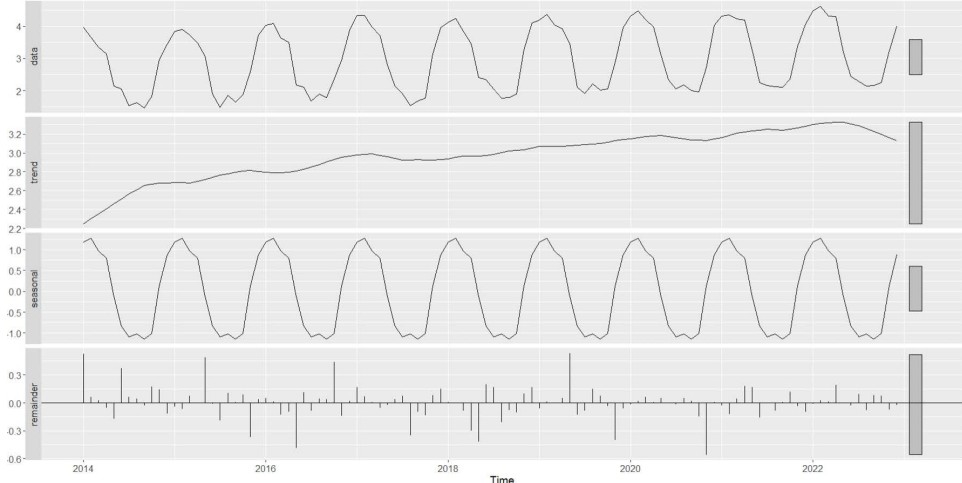

**Fig 2. The STL decomposition of the logarithm of Kayseri's corrected NG consumption data (January 2014 to December 2022).**

## 4. Results and discussion

### 4.1. Correction of NGD data during summer season

For the correction of NGD values in these months, a linear model is proposed:

$$D_{y,m} = T_{y,m} + R_{y,m} + e_{y,m}$$

$$(23)$$

Here, $D_{y,m}$ denotes the demand observed in the $y^{th}$ year and $m^{th}$ month; $T_{y,m}$ is the long-term effect on demand, to be extracted from the annual trend for the $m^{th}$ month. The second additive term, which is assumed to be orthogonal to the first, is $R_{y,m}$, which denotes the short-time effect, to be extracted from NGD in the recent months, and the last term is the unpredictable zero-mean error term. Trend estimates can be obtained by employing simple linear regression

on $D_{y,m}$ vs. year data for each month since a single global fit (to untransformed NGD) seems to be a poor fit for all the months (Fig 3A):

$$\hat{D}_{y,m} = T_{y,m} = \hat{\beta}_{0m} + \hat{\beta}_{1m}y \tag{24}$$

It is assumed that there exists (at least) lag-1 (one month) correlation between $D_{y,m}$ values for successive months, stemming from $R_{y,m}$ terms, i.e., $Corr(R_{y,m-1}, R_{y,m})$. Noting that $R_{y,m} + e_{y,m} = D_{y,m} - T_{y,m}$, and the random error terms are independent, the lag-1 correlation can be estimated from $Corr\ (D_{m-1} - T_{m-1}, D_m - T_m)$ using $D_m = [D_{1,m}\ D_{2,m}\ \cdots D_{9,m}]^T$ and $T_m = [T_{1,m}\ T_{2,m}\ \cdots T_{9,m}]^T$ for the $m^{th}$ month (see S1 Text for more information). In the light of the above modeling perspective, the correction procedure of NGD values during the summer season for six successive years, yielding $6 \times 3 = 18$,

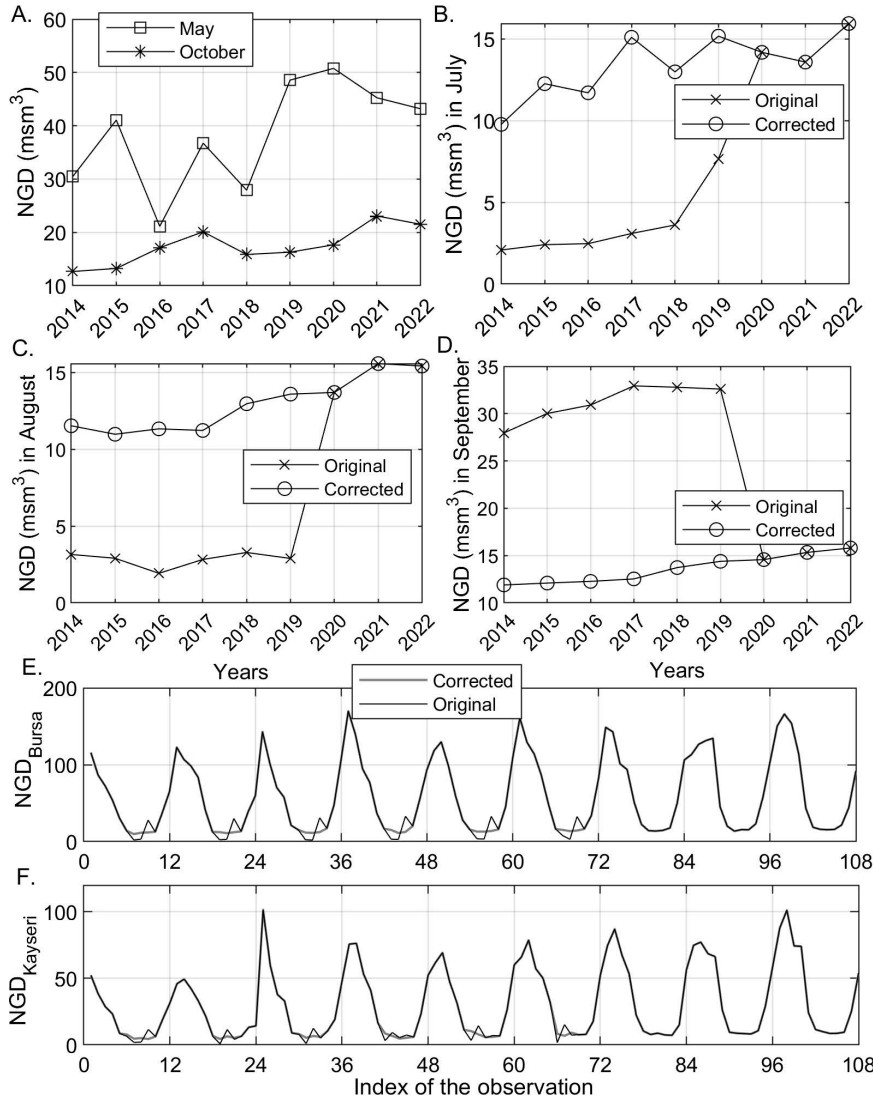

**Fig 3. (A) NGD values in Bursa for May and October over 2014-2022.** The original and corrected NGD values in Bursa for (B) July, (C) August, and (D) September 2014-2022. The original and corrected ("smoothed" gray lines during the summer periods in the figures) monthly NGD values in the training sets of (E) Bursa and (F) Kayseri.

and 6×4=24 data points for Bursa and Kayseri, respectively, is formulated as an optimization problem. Given monthly NGD data for nine years in $\boldsymbol{D}_m$, $m = 1, 2, ..12$, it is desired to replace $D_{y,m}$ with $y = 1, 2, ..6$ and $m = m_0, m_0 + 1, \ldots 9$ ($m_0$ is taken to be 7 and 6 for Bursa and Kayseri, respectively) with the optimum $\boldsymbol{D}_c = \{D^c_{y,m}; y = 1, 2, ..6; m = m_0, m_0 + 1, \ldots 9\}$ values ($c$ stands for corrected data), resulting in $\boldsymbol{D}^c_m = \begin{bmatrix} D^c_{1,m} & D^c_{2,m} \ldots & D^c_{6,m} & D_{7,m} & D_{8,m} & D_{9,m} \end{bmatrix}^T$ (note that the last three $D_{y,m}$

values are correct readings and are therefore treated as constants). Defining $\Phi = \begin{bmatrix} 1 & 1 \\ 1 & 2 \\ \vdots & \vdots \\ 1 & 9 \end{bmatrix}_{9 \times 2}$, the following objective

function is proposed for minimization (see S2 Text for details about the optimization function and the accompanying constraints):

$$\Psi(\boldsymbol{D}_c) = -\left\{ \sum_{m=m_0}^{9} \frac{corr(\boldsymbol{D}^c_m, \boldsymbol{T}_m)}{(9 - m_0 + 1)} + \sum_{m=m_0}^{10} \frac{\left| corr(\boldsymbol{D}^c_{m-1} - \boldsymbol{T}_{m-1}, \boldsymbol{D}^c_m - \boldsymbol{T}_m) \right|}{(10 - m_0 + 1)} \right\} \tag{25}$$

where $\boldsymbol{T}_m = \Phi \left( \Phi^T \Phi \right)^{-1} \Phi^T \boldsymbol{D}^c_m = \boldsymbol{H} \boldsymbol{D}^c_m$ for $m = m_0, m_0 + 1, \ldots 9$, and $\boldsymbol{T}_m = \boldsymbol{H} \boldsymbol{D}_m$ otherwise, subject to the following constraints:

$$\sum_{m=m_0}^{9} D^c_{y,m} = \sum_{m=m_0}^{9} D_{y,m}, \ y = 1, 2 \ldots 6 \tag{26}$$

$$D^c_{y,m} \geq 0, \ y = 1, 2, ..6; m = m_0, m_0 + 1, \ldots 9 \tag{27}$$

The optimization was performed with *fmincon* in the Optimization Function of MATLAB™ using the interior-point method [100] with the initial parameter values, i.e., the monthly NGD values during the summer period, all assumed to be equal to 10 msm³, slightly lower than the actual readings obtained during the last three years. The results of the optimization are shown in Fig 3B-D for Bursa. In addition, Figs 3E-F show that the irregular behavior during the summer season seen in the original time series has been corrected for both cities.

## 4.2. Time series modeling

Outliers were identified in both "corrected" datasets: five in the training set for Bursa and two for Kayseri. During the modeling process, cases in which these values were not smoothed and cases in which they were smoothed were treated separately. Both datasets indicated the utility of a logarithmic transformation; therefore, models were constructed separately for the original data and the transformed data. In light of the observed trend and seasonality in the data, we produced forecasts using four primary statistical models as a benchmark for the proposed approach: STL, SARIMA, ETS, and TBATS. We also implemented a machine learning-based combination technique, the AI-AFTER algorithm. The elapsed time for forecasting did not exceed 25 seconds in any case. The models were constructed using the training set, and the estimated parameters were subsequently applied to the validation data. As the performance of the models on the training and validation sets did not exhibit any anomalies, there was no necessity to update the parameters. All of the model orders were verified in several in-sample forecasting trials. Table 2 below provides descriptions of some of the models considered in this study. Once the model parameters were estimated using the training sample, 19-step-ahead forecasts were produced by the candidate models to evaluate their post-sample performances.

**Table 2. Details on the time series models applied to the logarithms of the corrected NG data.**

| Abbreviation | City | Description and References |
|---|---|---|
| ETS | Bursa | Additive Holt-Winters' Method with multiplicative errors and parameters $\alpha = 0.00113$, $\beta = 0.00010$, and $\gamma = 0.00011$. |
| | Kayseri | Additive Holt-Winters' Method with additive errors and parameters $\alpha = 0.00290$, $\beta = 0.00289$, and $\gamma = 0.00013$. |
| SARIMA | Bursa | Seasonal Autoregressive Integrated Moving Average model with drift, designated as $p = 1$, $d = 0$, $q = 0$; $P = 2$, $D = 1$, $Q = 0$. |
| | Kayseri | Seasonal Autoregressive Integrated Moving Average model with drift, designated as $p = 1$, $d = 0$, $q = 0$; $P = 0$, $D = 1$, $Q = 1$. |
| STL | Bursa | The data for both cities was subjected to additive seasonality with parameters s.window = "periodic" and t.window = 13, and robust fitting was adopted in the LOESS procedure. |
| | Kayseri | |
| TBATS | Bursa | Omega, the Box-Cox parameter = 0.91; ARMA(1,0) model was fitted; no damping; # of seasonal periods = 12; # of Fourier terms used for each seasonality = 4. |
| | Kayseri | Omega, the Box-Cox parameter = 1.00; ARMA(0,0) model was fitted; no damping; # of seasonal periods = 12; # of Fourier terms used for each seasonality = 5. |

The predictive performances of the time series models examined in this study are evaluated using the accuracy metrics provided in Equations 20–22. TBATS and the AI-AFTER algorithm exhibited superior performance in comparison to the other models for Bursa and Kayseri, respectively, as illustrated in Table 3. It must be noted that the initial 35 observations in the training set were used to train the candidate forecasts by the AI-AFTER algorithm, and were consequently not accessible to the analyst. Thus, the performance metrics of this method on the Kayseri training set were determined using a total of 73 observations, rather than 108.

Figs 4 and 5 illustrate the training and test data (black lines), and the predictions (blue lines) and forecasts (red dashed lines) generated by the TBATS model for Bursa and the AI-AFTER algorithm for Kayseri, respectively. Finally, Figs 6 and 7 demonstrate how well the standardized residuals from each of these two models fit to a proper normal distribution. While three of the residuals generated by the TBATS model are potential outliers, the number of such cases is two for the AI-AFTER algorithm. However, the Ljung-Box test statistic was found insignificant at lag 4 in both cases ($Q = 3.092$ for Bursa and $Q = 4.362$ for Kayseri), indicating that autocorrelation in the residuals is not an issue at $\alpha = 0.05$.

## 4.3. JITL-GPR modeling

The JITL-GPR forecasting method consists of two steps: (i) selecting a convenient subset of training set, i.e., the local training set, for each query point, and (ii) applying GPR to the local training set and predicting the query point.

**Table 3. Accuracy metrics for the benchmark time series model on the training and test sets.**

| City | Bursa | | | | | Kayseri | | | | |
|---|---|---|---|---|---|---|---|---|---|---|
| Model | SARIMA | ETS | STL | TBATS | AI-AFTER | SARIMA | ETS | STL | TBATS | AI-AFTER |
| $RMSE_{1\text{-step}}$ (msm³) | 9.212 | 9.569 | 9.366 | 9.024 | 8.863 | 5.383 | 5.120 | 5.956 | 5.221 | 5.275 |
| $MAE_{1\text{-step}}$ (msm³) | 5.917 | 6.185 | 6.300 | 6.355 | 6.023 | 3.593 | 3.409 | 3.888 | 3.446 | 3.657 |
| $MAPE_{1\text{-step}}$ (%) | 10.43 | 10.02 | 11.40 | 11.08 | 9.39 | 13.21 | 13.53 | 16.86 | 13.19 | 12.59 |
| $RMSE_{test}$ (msm³) | 19.221 | 20.361 | 19.566 | **14.857** | 16.751 | 10.778 | 8.286 | 9.013 | 6.786 | **6.332** |
| $MAE_{test}$ (msm³) | 14.367 | 13.244 | 14.211 | **9.666** | 11.208 | 7.822 | 5.828 | 7.189 | 5.094 | **4.140** |
| $MAPE_{test}$ (%) | 19.79 | 18.34 | 20.74 | **14.24** | 15.60 | 18.21 | 14.62 | 20.12 | 13.26 | **11.87** |
| NGD in 2023 (msm³) | 816.8 | 816.8 | 816.8 | 816.8 | 816.8 | 477.6 | 477.6 | 477.6 | 477.6 | 477.6 |
| NGD in 2023 prediction (msm³) | 850.9 | 867.7 | 690.8 | 779.3 | 797.9 | 519.7 | 502.1 | 401.6 | 451.0 | 475.4 |
| Yearly PE in 2023 | 4.18 | 6.23 | −15.42 | −4.58 | −2.31 | 8.83 | 5.15 | −15.90 | −5.55 | −0.45 |

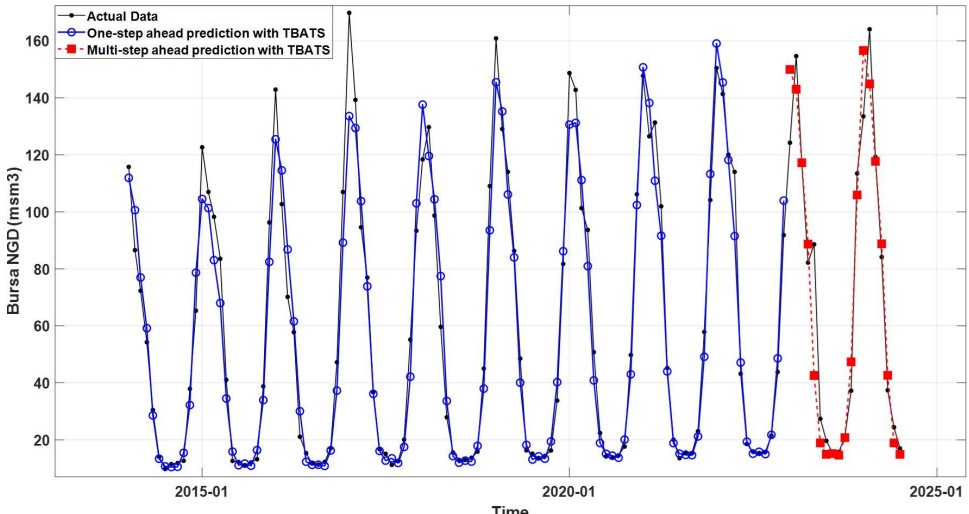

**Fig 4. The real NGD values, one-step ahead training set predictions, and multi-step ahead test set predictions using the TBATS model for Bursa.**

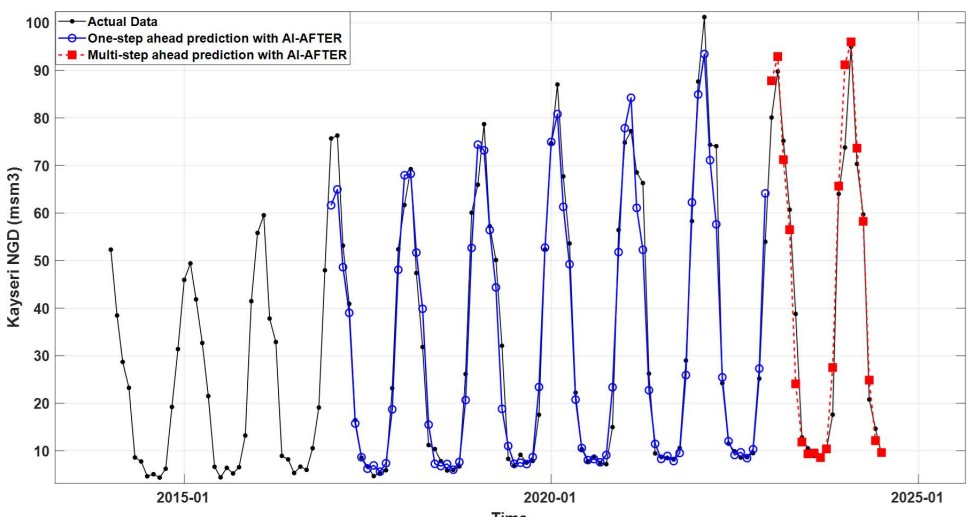

**Fig 5. The real NGD values, one-step ahead training set predictions, and multi-step ahead test set predictions using the AI-AFTER algorithm for Kayseri.**

**4.3.1. Local subset selection.** In the traditional time series representation for regression in NGDP, including the studies using machine learning tools, the features (excluding exogenous variables) correspond to the response variable sampled at previous time lags [23,59,101]:

$$\hat{D}_{t+1} = f(\boldsymbol{x}_t), \quad \boldsymbol{x}_t = [D_t \; D_{t-1} \ldots \; D_{t-p+1}]^T \in \mathbb{R}^p \tag{28}$$

In the present study, however, a different feature representation is adopted. The input features are assumed to be year and month values, i.e., $y = 1, \; 2, \; ..n_T; m = 1, 2, \ldots 12$, and for each point on this two-dimensional (2-D) grid, similar to a

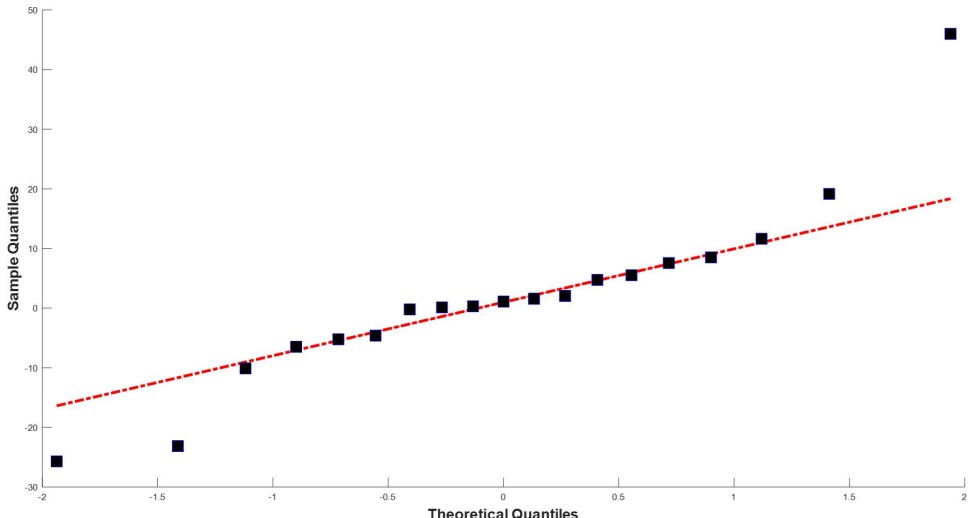

**Fig 6. QQ Plot for the multi-step ahead test set prediction residuals for Bursa.**

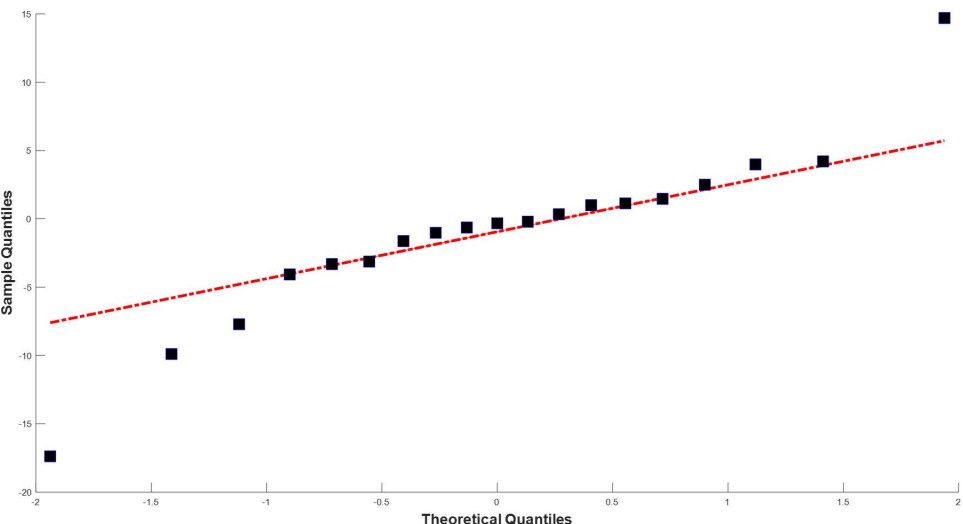

**Fig 7. QQ Plot for the multi-step ahead test set prediction residuals for Kayseri.**

2-D Gaussian random field, the NGD value $D_{y,m}$ is taken to be the response variable. Thus, the training (historical) dataset is defined as $TS = \{(y, m, D_{y,m}) ; y = 1, 2, ..n_T; m = 1, 2, \dots 12\}$. To predict the future query points (test set), a data segment with convenient window widths ($W_y$, $W_m$), representing the number of previous years and months, respectively, is extracted from the historical dataset, and GPR is employed on this 2-D data window. This procedure is to be repeated for all query points, thereby rendering the proposed method an application of JITL. It is to be noted that the similarity of the query point to the training data is based on proximity in years and months (on the 2-D grid), and is computed using modular arithmetic. In other words, the month indices 1 (January) and 12 (December) in two successive years are assumed to be in close proximity, similar to the rationale in time series analysis. Thus, for a query point with the index $q = 12y + m$, in which $y$ and $m$ values are known, the following function $\Phi\left(T, q, W_y, W_m\right)$ is used to extract the relevant subset from

the training dataset $TS$ with the help of an accompanying function $f(n)$, which extracts year, month and NGD value of the $(n-1)^{th}$ observation via $f(n) = (ceil((n-1)/12),\ mod(n-2,12)+1,\ D(n-1))$:

$$\Phi\left(T, q, W_y, W_m\right) = \{f(n) = (y, m, D_{y,m}),\ n \in \{1, 2, ..q\}\} = \{f(q), f(q-1), \dots, f(q-W_m+1), f(q-11),$$
$$f(q-12), \dots, f(q-11-(W_m-1)), \dots f(q-(n \times 12-1)), f(q-n \times 12), \dots f(q-(n \times 12-1-(W_m-1))), \dots,$$
$$f\left(q-\left((W_y-1) \times 12-1\right)\right), \dots f\left(q-\left((W_y-1) \times 12-1-(W_m+1)\right)\right)\}$$

(29)

In the equation above, *ceil*() and *mod*() are ceiling and mod functions, respectively. While a quick look at Equation 29 gives the impression of a rather complicated local dataset construction method, it is indeed quite simple. As an example, the construction of the local training set for the first query point in the test set (the 109th point, January 2023) is demonstrated for $W_y$=4 and $W_m$=3. Table 4 shows the years and months of the data in the TS in 2-D grid form, and NGD values (in time-index format) are placed in each corresponding cell. The query point in the 11th year is shown with a dark gray cell, where $D_q$ is unknown. Since $W_m$=3, the last three months, including the query month, are included in the local training set, hence $D_{106}$ and $D_{107}$, corresponding to November and December of 2022, are selected (in light gray color). For one-year lagged observations, NGD values for November 2021, December 2021, and January 2022 are selected, corresponding to $D_{95}$, $D_{96,}$ and $D_{97}$, respectively. This selection method is repeated to include observations up to $W_y$ years, which corresponds to the selection method proposed in Equation 29. In this way, the historical samples that are similar to the current query point in terms of the seasons are selected. To construct the local training set for the next query point (110th point, February 2024), the same procedure is repeated using the $W_y$ and $W_m$ values for February, while the already predicted value for January is used as a training set point. Thus, multi-step-ahead forecasting is achieved in an iterative manner.

The $k^{th}$ element of the $(y, m)$ pair in the subset extracted by $\Phi\left(T, q, W_y, W_m\right)$ is denoted by $\boldsymbol{x}_k$, i.e., $\Phi\left(T, q, W_y, W_m\right) = \{(\boldsymbol{x}_k, D_k),\ k = \{1, 2, ..W_y \times W_m - 1\}\}$; furthermore, $\boldsymbol{X}_{JITL} = \{\boldsymbol{x}_k,\ k = \{1, 2, ..W_y \times W_m - 1\}\}$ and $\boldsymbol{D}_{JITL} = \{D_k,\ k = \{1, 2, ..W_y \times W_m - 1\}\}$ are used for ease of representation. For a given query point $\boldsymbol{x}_q$, $\boldsymbol{X}_{JITL}$ and $\boldsymbol{D}_{JITL}$ are determined and z-score scaled. Finally, GPR is used (see Equation 18) to determine the expected value of NGD at the query point, $E\left\{D_q \mid x_q, \boldsymbol{X}, \boldsymbol{D}_{JITL}\right\} = f\left(x_q \mid \boldsymbol{X}_{JITL}, \boldsymbol{D}_{JITL}\right)$ (please note that the symbol $y$ used in Sections 2.1 and 2.2 for the output variable is now reserved for the feature year number) using the *fitgrp* function in the Statistics and Machine Learning Toolbox of MATLAB™.

**4.3.2. Kernel design.** Based on the observation that NGD values exhibit a nearly linear trend in the long-run, with sinusoidal-like behavior within each year, an additive kernel that consists of a homogeneous linear kernel for years and a second order polynomial kernel for months is designed. Given two vectors $\boldsymbol{x}_u = [y_u\ m_u]^T$ and $\boldsymbol{x}_v = [y_v\ m_v]^T$, with $u, v \in \{1, 2, \dots\}$, index of the observation, two basic kernels are added to form a convenient kernel function.

$$k\left(\boldsymbol{x}_u, \boldsymbol{x}_v\right) = \sigma_f^2 \left(\beta^2 y_u^T y_v + \left(m_u^T m_v + \alpha^2\right)^2\right)$$

(30)

**Table 4. An example of how a local training set is selected in JITL-GPR model.**

| Years/ Months | 1 | 2 | ...... | 7 | 8 | 9 | 10 | 11 |
|---|---|---|---|---|---|---|---|---|
| 1 | $D_1$ | $D_{13}$ | … | $D_{73}$ | $D_{85}$ | $D_{85}$ | $D_{97}$ | $D_q = D_{109}$ |
| 2 | $D_2$ | $D_{14}$ | … | $D_{74}$ | $D_{86}$ | $D_{86}$ | $D_{98}$ | |
| … | … | … | … | … | … | … | … | … |
| 10 | $D_{10}$ | $D_{22}$ | … | $D_{82}$ | $D_{94}$ | $D_{94}$ | $D_{106}$ | |
| 11 | $D_{11}$ | $D_{23}$ | … | $D_{83}$ | $D_{95}$ | $D_{95}$ | $D_{107}$ | |
| 12 | $D_{12}$ | $D_{24}$ | … | $D_{84}$ | $D_{96}$ | $D_{96}$ | $D_{108}$ | |

The three hyperparameters in GPR, $\sigma_f^2$, $\beta^2$, and $\alpha^2$, are expected to adjust the variance of the response variable, the weight of the "year effect" relative to the "month effect" (or scale parameter), and the curvature of the NGD in recent months. These three hyperparameters were all initialized with unity in the optimization of marginal loglikelihood, since all variables were z-normalized, and identical results were obtained using initial values between 0.2 and 5.0. Finally, the mean of the GPR function (see Equation 14) was set equal to zero, i.e., no fixed basis functions were used.

**4.3.3. Window size tuning.** To adjust the hyper-parameters of the JITL-GPR model, the first 48 observations in the TS were reserved as a "buffer" region, and observations with indices 49–108 (60 data points) in the TS were predicted using a moving horizon estimation method, i.e., once a query point is predicted, it is included in the training set accessible for the next query sample. Here, a grid search over the local dataset window sizes of $W_y = \{2, 3, \ldots 8\}$ and $W_m = \{2, 3, \ldots 6\}$ (see S3 Text for the rationale of selecting these ranges for the hyperparameters) is performed to find a single $(W_y, W_m)$ parameter pair (applicable to all query points) with the smallest one-step-ahead RMSE value. Using this method on Bursa yielded an RMSE of 14.1 msm³ at $(W_y, W_m) = (8, 3)$; however, a surface plot of RMSE values over all the parameter value combinations showed that RMSE≤15.5 msm³ was obtained for all $W_m$ values and $W_y \geq 4$. This indicated that the prediction accuracy did not change with respect to the number of previous months included in the model as long as more than four successive years of data were included. This was indeed unexpected; hence, we examined the phenomena in more detail. We grouped 60 predictions into 12 groups of five observations (years) each, and found that the RMSE surfaces with respect to parameter values changed significantly from month to month. This indicated that the flat RMSE surface with respect to window sizes is due to an "averaging out" effect over 12-months, i.e., using the same window size for all months yields an inferior and almost constant performance over a range of window sizes.

Due to the limited number of historical observations, with each month having only five observations available for tuning the window size, it is not feasible to tune a separate $(W_y, W_m)$ pair for each month. Combining "similar" months into single units is a better strategy, which has been frequently employed in the NGDP literature for predictive purposes [5,9]. To this end, the similarity of RMSE surfaces with respect to parameter values for all pairs of successive months was checked and series of similar months were grouped together (see S4 Text). Consequently, all NGD values during January-May (five months), June-September (four months), and October-December (three months) are grouped as Group I, Group II and Group III, respectively.

**4.3.4. Predictions of JITL-GPR for the test set.** For each group, which now contained 25, 20, and 15 observations, $(W_y, W_m)$ parameter values that yield the minimum RMSE, were found to be equal to 13.22 msm³ (~1 msm³ smaller than the previous value), were determined. The results for the two cities are shown in Table 5. It is interesting to note that the optimum values of the window parameters show a striking similarity for both cities, indicating a more general NGD tendency that is not localized to a specific location. The optimal value of $W_y$ varies less between the groups compared to that of $W_m$; only the NGD at the beginning of winter (Group III) seems to be less affected by the slow annual trend, but related to the very recent demand values. The information of the previous months was found to be more important for the months with higher heating demand. We can speculate that this information could be helpful for the GPR model in determining the recent trend in weather and economic conditions that affect the heating habits of the residents.

Two examples of multi-step-ahead predictions for the test set of Kayseri are demonstrated in Figs 8A and 8B for January (one-step-ahead) and November (11-step-ahead) 2023, respectively. Note that $\{W_y, W_m\}$ pair is taken to be equal

**Table 5. $W_y, W_m$ values that yield the minimum RMSE values of one-step ahead predictions.**

| City | Bursa | | | Kayseri | | |
|---|---|---|---|---|---|---|
| Groups | I | II | III | I | II | III |
| Optimum $W_y$ | 8 | 8 | 5 | 8 | 8 | 2 |
| Optimum $W_m$ | 6 | 2 | 4 | 6 | 2 | 4 |

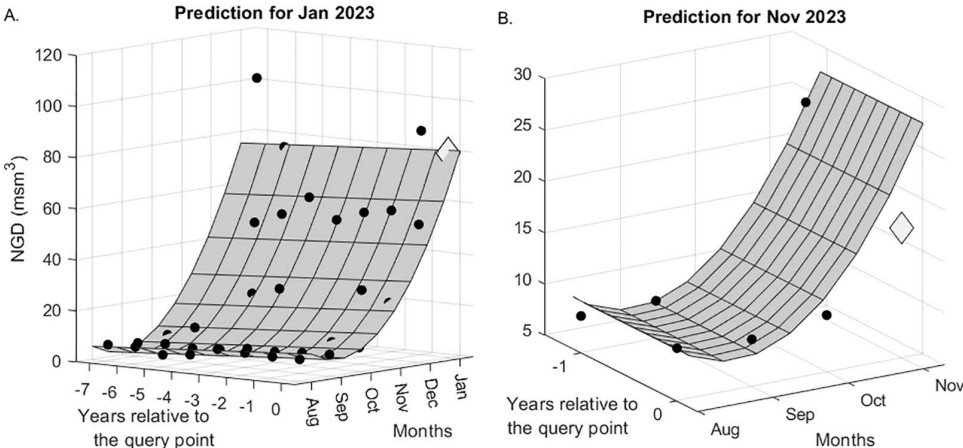

**Fig 8. NGDP for (A) January, and (B) November 2023.** The filled circles represent the training sets in both subfigures, and also NGDPs for August-October 2023 (in the left subfigure); while the large diamonds denote the true value of the query point.

to {8, 6} for January, and {2, 4} for November (see Table 5). Here, the $0^{th}$ year corresponds to the query time and the $W_m-1$ months before it, so the year is not taken as an absolute time measure. In both figures, the GPR surfaces fit the historical data well, and predict the test point quite accurately. Table 6 shows that the accuracy of the JITL-GPR predictions for the test set exceeds that of the time-series models; moreover, the percentage error of the prediction of the total NGD in 2023 is less than 1% for both cities, indicating that the government regulations for annual predictions are satisfactorily met. A comparison of the results of JITL-GPR with those from time series methods, as presented in Section 4.2, indicates that the JITL-GPR approach led to a reduction in out-of-sample RMSE values by 14.6% for Bursa and 19.3% for Kayseri, as well as a decrease in MAE values by 2.5% for Bursa and 10.4% for Kayseri (see Fig 9). The proposed model led to a 5.1% reduction in MAPE for Bursa, yet it did not achieve the same outcome for Kayseri. Fig 10 shows one-step ahead predictions for the training set and the multi-step ahead predictions for the test set for both cities, again indicating that the forecasts adequately follow the actual demand values. Finally, the prediction residuals, defined as the difference between the predicted and actual NGD values, generally show a similar picture for both cities (see Fig 11). Overpredicted NGD values in the first year may be due to small number of historical data (only the first four years of data

**Table 6. Accuracy metrics for the JITL-GPR and TBATS models on the training and test sets.**

| City | Bursa | | Kayseri | |
|---|---|---|---|---|
| **Model** | **JITL-GPR** | **TBATS** | **JITL-GPR** | **TBATS** |
| $RMSE_{1-step}$ (msm³) | 13.23 | 9.02 | 6.407 | 5.221 |
| $MAE_{1-step}$ (msm³) | 8.485 | 6.355 | 4.590 | 3.446 |
| $MAPE_{1-step}$ (%) | 12.20 | 11.08 | 16.34 | 13.19 |
| $RMSE_{test}$ (msm³) | **12.69** | 14.86 | **5.110** | 6.786 |
| $MAE_{test}$ (msm³) | **9.424**[i] | 9.666 | **3.711** | 5.094 |
| $MAPE_{test}$ (%) | **13.51** | 14.24 | 14.63 | **13.26** |
| NGD in 2023 (msm³) | 816.8 | 816.8 | 477.6 | 477.6 |
| NGD in 2023 prediction (msm³) | **824.3** | 779.3 | **473.6** | 451.0 |
| Yearly PE in 2023 | **0.90** | −4.58 | **−0.83** | −5.55 |

[i]Higher test performances for both cities are shown in bold.

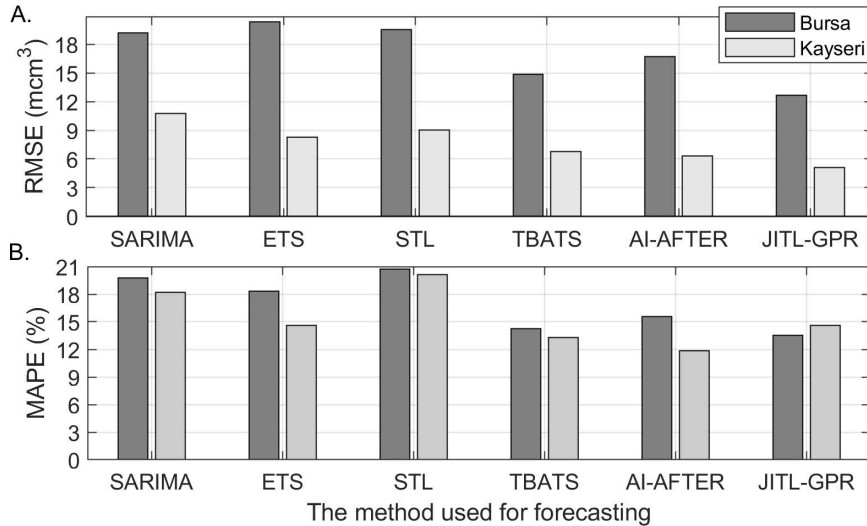

**Fig 9. Comparison of RMSE and MAPE values of different methods for the test set.**

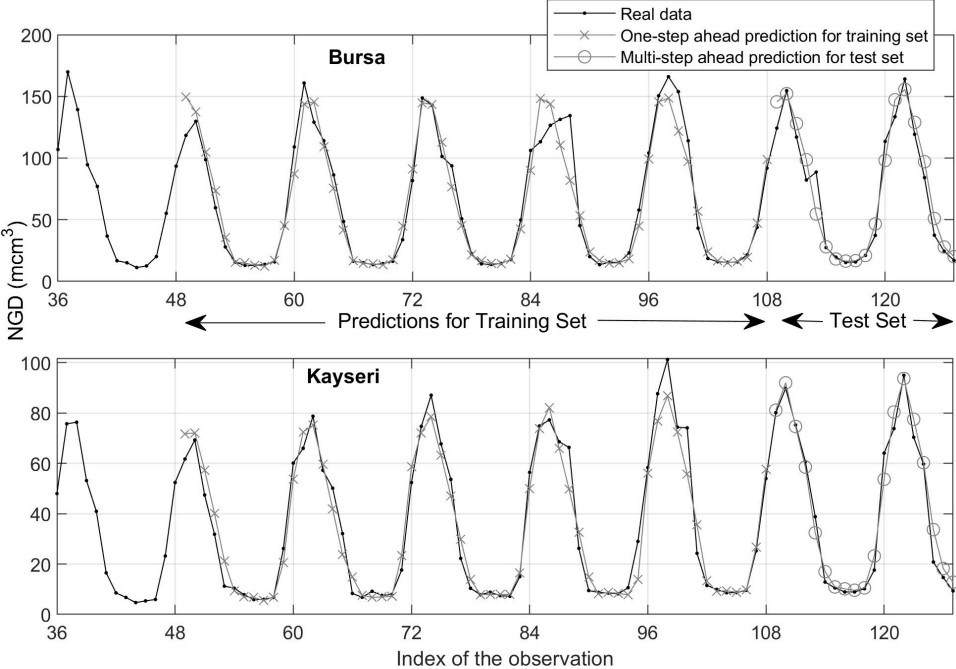

**Fig 10. Comparison of real, one-step ahead training set and multi-step ahead test set predictions of NGD values.**

are available to predict the 49th observation), especially to accurately determine the annual trend, while underpredicted NGD values in the winter of 2021–2022 (observations #95–100) may be due to a concept drift, such as economic conditions imposed by the COVID pandemic. However, the advantage of using an adaptive JITL mechanism is seen here; the inclusion of these "unexpected" NGD values within the window of the JITL-GPR leads to more accurate predictions of the next year's NGDPs.

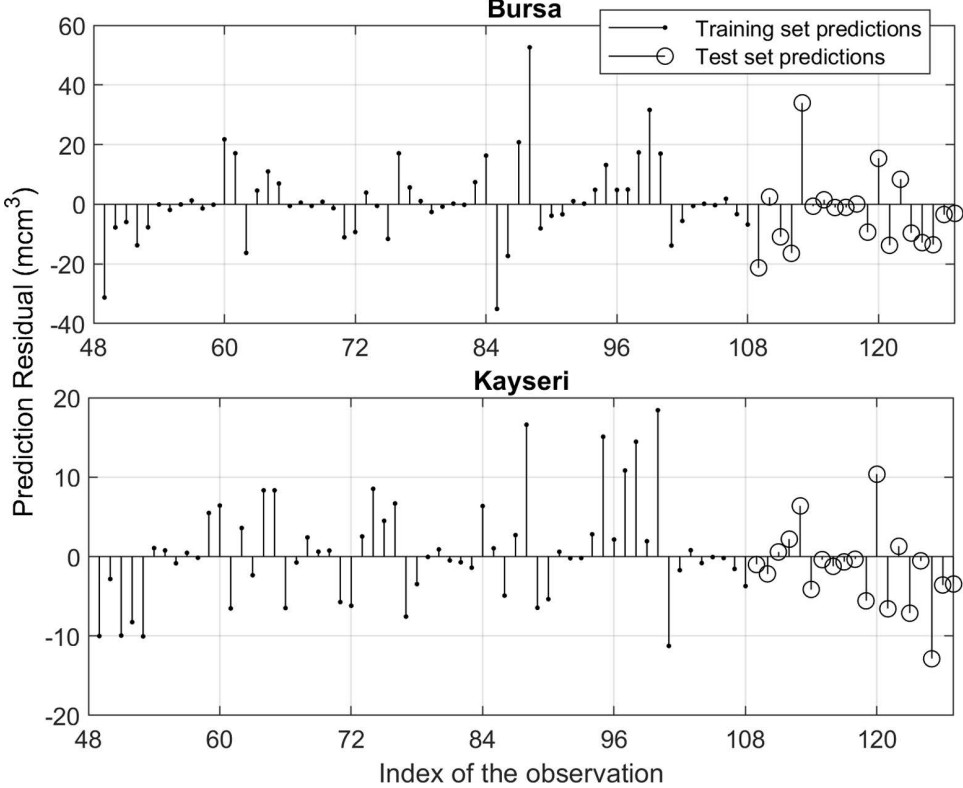

**Fig 11. One-step ahead training set and multi-step ahead test set prediction residuals.**

**4.3.5. Application of JITL-GPR on an additional dataset of NGD in Turkey.** To show the efficacy of the proposed method in achieving high forecast accuracy for other datasets, an additional demostration was conducted on the NGD values of Turkey (the entire country) for a 129-month period between 2010 and 2020. The data for this experiment was obtained from https://www.kaggle.com/datasets/naturalgas/turkeys-monthly-natural-gas-data). In a manner analogous to the partitioining strategy implemented for Bursa and Kayseri, the initial 108 months (2010–2018 in the supplementary dataset) were allocated for training, while the subsequent 21 months (2019–2020) were designated as the test (held-out) set. We employed two methods, JITL-GPR and SARIMA, on the additional dataset. To circumvent selection bias in the combination of similar months for the JITL-GPR method, the identical month groupings that were employed for Bursa and Kayseri (see S4 Text) were utilized. Moreover, the identical hyperparameter search space and kernel function were employed to ascertain the hyperparameters with the minimal validation errors. The SARIMA model that yielded the smallest AIC value, with reasonable residual distributions and autocorrelations, was observed to generate a subtantial test error. Consequently, a SARIMA model was sought that yielded the smallest RMSE on the test error via a trial-and-error process. It should be stressed that, subsequent to the determination of posteriori best test errors from SARIMA, an unwarranted advantage was ascribed to this method in comparison to the proposed JITL-GPR method. Conversely, the results show that the accuracy of the unbiased multi-step ahead predictions of JITL-GPR remains superior to the accuracy of the (biased) predictions by SARIMA (see Table 7). Addionally, the MAPE value obtained from JITL-GPR is approximately 15%, a value that is compatable to those observed for Bursa and Kayseri. This suggests that the accuracy metrics presented in this study are likely to be unbiased estimations of future performance.

**Table 7. Accuracy metrics for the JITL-GPR and SARIMA models on the additional dataset.**

|  | Model | |
| --- | --- | --- |
|  | **JITL-GPR** | **SARIMA** |
| $RMSE_{1-step}$ (msm$^3$) | 378.3 | 461.0 |
| $MAPE_{1-step}$ (%) | 7.36 | 9.36 |
| $RMSE_{test}$ (msm$^3$) | **568.1** | 691.5 |
| $MAPE_{test}$ (%) | **15.1** | 20.6 |

As illustrated in Fig 12, the time-series plots of the NGD values for the 129-month period in Turkey further substantiate the efficacy of the JITL-GPR method in predicting outcomes. For instance, the maximum NGD in January 2020 (13th test data) is well predicted by JITL-GPR, while underpredicted by SARIMA, demonstrating the efficacy of the former in predicting extreme values, particularly during winter.

## 5. Conclusion and recommendations

NG has emerged as a pivotal source of energy, exhibiting a nearly constant yearly increase in worldwide demand over the past ~30 years. The reliance of numerous countries, including Turkey, on imports to maintain a sufficient NG supply underscores the significance of accurate NGDP in economic planning by governments. The present study aims to construct an ex-ante predictive model (rather than an explanatory model) that can be used under realistic conditions. The monthly NGDPs of two Turkish cities were examined employing various time series models and a novel online learning

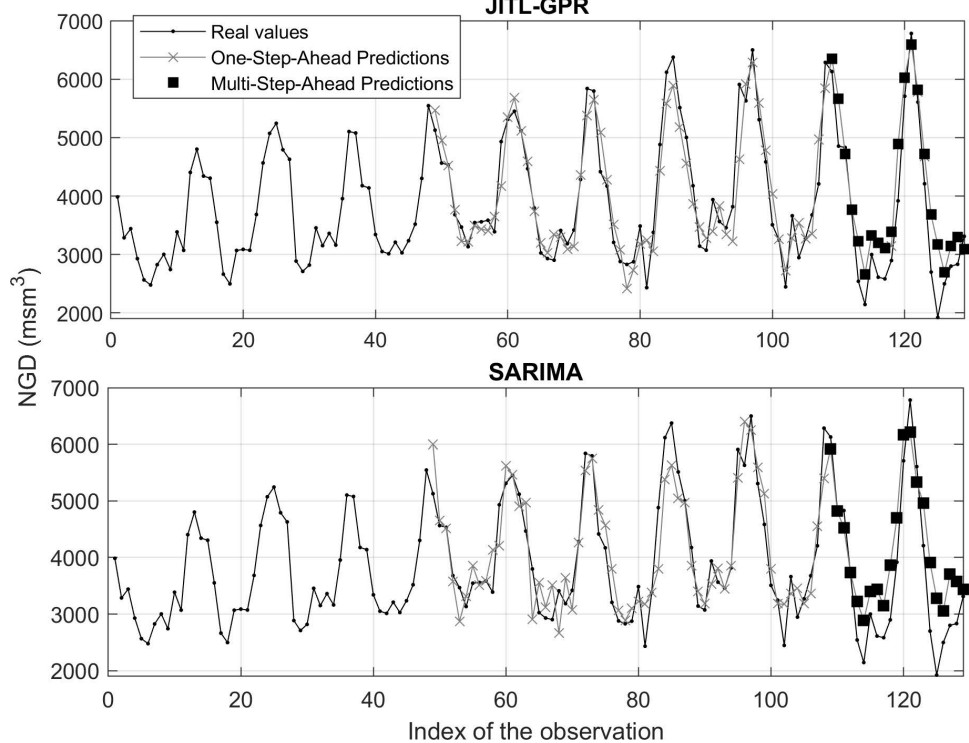

**Fig 12. One-step ahead training set and multi-step ahead test set predictions for NGD values for Turkey.**

method, JITL-GPR. Moreover, the JITL-GPR model demonstrated remarkable precision in forecasting the yearly NGD for 2023, achieving an out-of-sample prediction error of less than 1% for both Bursa and Kayseri.

The current methodology employs a weighting system that considers the proximity of previous NGD values to the query point in months and years, in contrast to the utilization of each previous NGD value as a separate feature, i.e., in a separate column of a regressor matrix. This methodology then adjusts the weights to achieve the best fitting (in Bayesian sense) via GPR (see Equation 18). The proposed approach bears resemblance to the "kriging" method employed in geostatistics, wherein Gaussian processes are used in low-dimensional spatial dimensional settings [102]. The proposed method can be regarded as a regularized version of the standard prediction methods, wherein the input features are derived from lagged output values. In contrast to the SARIMAX method, which seeks to maximize the likelihood under the constraints of stability and invertibility, the proposed method employs a regularization approach that restricts the impact of NGD values from previous years and months on the current year. This ensures proximity in year and month (weighted) directions, thereby adapting to changing conditions. It has been demonstrated that JITL-GPR exhibits a propensity to resist outliers, a phenomenon that can be attributed to the smoothing-out effect of the low-order kernel.

Despite the geographical separation of Bursa and Kayseri, which is approximately 660 km, both have Mediterranean and continental climates. However, comparable optimal window parameters for GPR-JITL were obtained in both locations. This agreement could be due to two possible factors: (i) an artifact resulting from the approximation used in aggregating the months from January to May into a single category to address the issue of inadequate sample size, or (ii) medium-term (monthly) NGD values, which exhibit reduced sensitivity to weather conditions, compared to social variables affecting a broader community or nation. This observation, in conjunction with the selection of window parameters during the winter months, requires further investigation. Despite the fact that the proposed approach is applicable to any monthly NGD data comprising 50–200 observations, there is still room for improvement with regard to real-world forecasting. The proposed method for aggregation of months, which involves the partitioning of a single year into three periods (January-May, June-September, and October-December), and the proposed kernel function have demonstrated effectiveness in the tested datasets. However, there is potential for enhancement of both the aggregation process and the kernel function in the future, with the objective of aligning with the NGD values derived from a more extensive range of geographic and socioeconomic conditions. In addition, the autocorrelation structure of the prediction residuals of GPR-JITL suggests that the proposed method may require further refinement to adapt to the rapid fluctuations in NGD, such as abrupt drifts. This may require the incorporation of recent observations, for instance, using weighted GPR [103], or the integration of predictions with time series models within a transfer learning framework [66]. The proposed approach was also applied to monthly natural gas consumption per subscriber, but the results did not indicate a significant improvement. Furthermore, such a model would require the prediction of the number of subscribers in the coming months, which would introduce additional uncertainty. Therefore, only the current results were reported. Finally, the incorporation of "Ramadan effect" [104] into the JITL-GPR framework, if it exists, could potentially improve the accuracy of NGDP values. Further research is necessary to investigate these aspects and to improve the prediction accuracy of NGD values.

## Supporting information

**S1 Text. Correlation of the short-term effect in two successive months.**
(DOCX)

**S2 Text. Explanation of optimization function and its constraints.**
(DOCX)

**S3 Text. The rationale for using the specified ranges of window sizes for grid search.**
(DOCX)

**S4 Text. Window size tuning for different months in JITL-GPR method.**
(DOCX)

**S5 Appendix. Summary of NGDM studies in Turkey.**
(DOCX)

## Acknowledgments

The authors would like to express their deep gratitude to Şükran Sibel MENTEŞ, Deniz DEMİRHAN, and Hakan GÜL for their exemplary guidance and expertise, which significantly increased the caliber and scope of this study. The dataset provided by SOCAR proved to be indispensable for the advancement of this study. Therefore, the authors are grateful to Fuad İBRAHİMOV (project sponsor), Shamil GULUZADE, and Ahmet YİĞİT for their cooperation in the use of the research data and leadership in the successful completion of this research. In addition, we express our deep gratitude to Baran SAVCI, Nihal BÜTÜN, Nazlı ACAR, Firdevs Betül HAKYEMEZ, Ceyhun BAHTİYAR, and İsmail ULU from SOCAR for their unwavering support and assistance throughout the research process.

## Author contributions

**Conceptualization:** Burak Alakent, Erkan Isikli, Cigdem Kadaifci.

**Data curation:** Burak Alakent, Erkan Isikli, Cigdem Kadaifci.

**Formal analysis:** Burak Alakent, Erkan Isikli.

**Methodology:** Burak Alakent, Erkan Isikli, Cigdem Kadaifci.

**Project administration:** Tonguc S. Taspinar.

**Resources:** Tonguc S. Taspinar.

**Supervision:** Tonguc S. Taspinar.

**Visualization:** Burak Alakent, Erkan Isikli, Cigdem Kadaifci.

**Writing – original draft:** Burak Alakent, Erkan Isikli, Cigdem Kadaifci.

**Writing – review & editing:** Erkan Isikli.

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
