## [Decision Letter · Decision Letter 0]

Dear Dr. Isikli,

Thank you for submitting your manuscript to PLOS ONE. After careful consideration, we feel that it has merit but does not fully meet PLOS ONE’s publication criteria as it currently stands. Therefore, we invite you to submit a revised version of the manuscript that addresses the points raised during the review process.

We look forward to receiving your revised manuscript.

Kind regards,

Justyna Żywiołek

Academic Editor

PLOS ONE

2. Please note that PLOS ONE has specific guidelines on code sharing for submissions in which author-generated code underpins the findings in the manuscript. In these cases, we expect all author-generated code to be made available without restrictions upon publication of the work. Please review our guidelines at https://journals.pl

Reviewers' comments:

Reviewer's Responses to Questions

**Comments to the Author**

1. Is the manuscript technically sound, and do the data support the conclusions?

Reviewer #1: Yes

2. Has the statistical analysis been performed appropriately and rigorously?

Reviewer #1: I Don't Know

3. Have the authors made all data underlying the findings in their manuscript fully available?

Reviewer #1: Yes

4. Is the manuscript presented in an intelligible fashion and written in standard English?

Reviewer #1: Yes

Reviewer #1: 1- Does the paper address the challenges of implementing JITL-GPR in real-world forecasting scenarios?

2- Does the proposed JITL-GPR method provide improved forecasting accuracy compared to traditional models such as SARIMA and ETS?

3- To what extent do external factors (e.g., temperature, gas price, and number of subscribers) influence forecasting accuracy?

4- Why are monthly forecasts less accurate compared to daily or yearly predictions?

5- Can the JITL-GPR method be generalized to other cities or countries that rely on imported natural gas?

6- Most of the figures are not clear, please.

**Do you want your identity to be public for this peer review?** For information about this choice, including consent withdrawal, please see our Privacy Policy

Reviewer #1: No

---

## [Author Response · Author response to Decision Letter 1]

9 Apr 2025

In light of your recommendations, several sentences have been incorporated into the conclusion section, and Table 6 has been revised. The proposed approach has also been applied to an openly accessible dataset to illustrate its efficacy. The findings have been disseminated exclusively to you, with the understanding that including them in the manuscript would result in an excessive accumulation of information. The images contained in the initial draft have been substituted with their high-resolution versions. For a more detailed overview of the changes made to the manuscript, please refer to the "Response to Reviewers" section.

---

## [Editor Report · Decision Letter 1]

Dear Dr. Isikli,

Thank you for submitting your manuscript to PLOS ONE. After careful consideration, we feel that it has merit but does not fully meet PLOS ONE’s publication criteria as it currently stands. Therefore, we invite you to submit a revised version of the manuscript that addresses the points raised during the review process.

**ACADEMIC EDITOR:** <h3 data-end="240" data-start="211">Changes that were made:</h3>

In the **"Response to Reviewers"** section, they addressed specific reviewer comments:

The **conclusion section was revised** , including new sentences that discuss limitations of applying the JITL-GPR method in practice.

**Table 6 was updated** , now including a comparison between the proposed JITL-GPR method and the TBATS model.

**All figures were replaced with high-resolution versions**

The authors **justified their decision not to include exogenous variables** (e.g., temperature, gas price, number of subscribers), stating that such data are difficult to forecast reliably at a monthly horizon and would introduce unnecessary uncertainty

The **conclusions were expanded** , for example with the sentence:

"Despite the fact that the proposed approach is applicable to any monthly NGD data comprising 50 to 200 observations, it may encounter certain challenges when implemented in real-world forecasting"

The **JITL-GPR method was applied to an additional publicly available dataset** (Turkey, 2010–2020), but the results were **not included in the manuscript** —they were shared only with the reviewers to avoid "excessive accumulation of information"

<h3 data-end="1661" data-start="1619">What was **not changed** or omitted:</h3>

The authors **did not include the results for the additional dataset in the manuscript** , explaining it would make the paper overly long.They did **not implement automated selection of similar months** for local model training, though they acknowledged it as a promising future direction to mitigate "selection bias".

We look forward to receiving your revised manuscript.

Kind regards,

Justyna Żywiołek

Academic Editor

PLOS ONE

Journal Requirements:

Additional Editor Comments:

Changes that were made:

In the "Response to Reviewers" section, they addressed specific reviewer comments:

The conclusion section was revised, including new sentences that discuss limitations of applying the JITL-GPR method in practice.

Table 6 was updated, now including a comparison between the proposed JITL-GPR method and the TBATS model.

All figures were replaced with high-resolution versions.

The authors justified their decision not to include exogenous variables (e.g., temperature, gas price, number of subscribers), stating that such data are difficult to forecast reliably at a monthly horizon and would introduce unnecessary uncertainty.

The conclusions were expanded, for example with the sentence:

"Despite the fact that the proposed approach is applicable to any monthly NGD data comprising 50 to 200 observations, it may encounter certain challenges when implemented in real-world forecasting".

The JITL-GPR method was applied to an additional publicly available dataset (Turkey, 2010–2020), but the results were not included in the manuscript—they were shared only with the reviewers to avoid "excessive accumulation of information".

What was not changed or omitted:

The authors did not include the results for the additional dataset in the manuscript, explaining it would make the paper overly long.

They did not implement automated selection of similar months for local model training, though they acknowledged it as a promising future direction to mitigate "selection bias".

---

## [Author Response · Author response to Decision Letter 2]

16 Apr 2025

In light of the recommendations provided by the reviewer, several sentences have been incorporated into the conclusion section, and Table 6 has been revised. The proposed approach has also been applied to an openly accessible dataset to illustrate its efficacy. The findings have been disseminated exclusively to the reviewer and the editor, with the understanding that including them in the manuscript would result in an excessive accumulation of information. The images contained in the initial draft have been substituted with their high-resolution versions. For a more detailed overview of the changes made to the manuscript, please refer to the "Response to Reviewers" section.

The full dataset, provided by SOCAR Türkiye, is not publicly available due to confidentiality restrictions. However, in accordance with your publication standards, the company has kindly consented to share a partial dataset of actual natural gas consumption for both cities considered in our study. This data can be accessed at https://github.com/Balakent/NGD-in-Two-Cities. Additionally, the figures presented in the manuscript also provide a clear representation of the data, and readers can refer to these for a comprehensive understanding of the results.

In light of the recommendations provided by the academic editor, a new section entitled “Application of JITLGPR on an additional dataset of NGD in Turkey” was incorporated into the manuscript. Two paragraphs were added to the manuscript to explain why selection bias is not a problem. A thorough explanation regarding this concern can be found in the "Response to Reviewers" document.

---

## [Editor Report · Decision Letter 2]

Forecasting Monthly Residential Natural Gas Demand in Two Cities of Turkey Using Just-In-Time-Learning Modeling

PONE-D-25-09289R2

Dear Dr. Isikli,

We’re pleased to inform you that your manuscript has been judged scientifically suitable for publication and will be formally accepted for publication once it meets all outstanding technical requirements.

Kind regards,

Justyna Żywiołek

Academic Editor

PLOS ONE
---

## [Editor Report · Acceptance letter]

PONE-D-25-09289R2

PLOS ONE

Dear Dr. Isikli,

I'm pleased to inform you that your manuscript has been deemed suitable for publication in PLOS ONE. Congratulations! Your manuscript is now being handed over to our production team.

Kind regards,

on behalf of

Dr. Justyna Żywiołek

Academic Editor

PLOS ONE